# Perinatal Outcomes and Level of Labour Difficulty in Deliveries with Right and Left Foetal Position—A Preliminary Study

**DOI:** 10.3390/healthcare12080864

**Published:** 2024-04-22

**Authors:** Magdalena Witkiewicz, Barbara Baranowska, Maria Węgrzynowska, Iwona Kiersnowska, Katarzyna Karzel, Grażyna Bączek, Dorota Sys, Anna Scholz, Susan Crowther, Justyna Teliga-Czajkowska, Urszula Tataj-Puzyna

**Affiliations:** 1Department of Gynecologic and Obstetrical Didactics, Medical University of Warsaw, 00-581 Warsaw, Poland; magdalena.witkiewicz@wum.edu.pl (M.W.); gbaczek@wum.edu.pl (G.B.); justyna.teliga-czajkowska@wum.edu.pl (J.T.-C.); 2Department of Midwifery, Centre of Postgraduate Medical Education, 01-004 Warsaw, Poland; maria.wegrzynowska@cmkp.edu.pl (M.W.); urszula.tataj-puzyna@cmkp.edu.pl (U.T.-P.); 3Department of Basic Nursing, Medical University of Warsaw, 01-445 Warsaw, Poland; ikiersnowska@gmail.com; 4Faculty of Psychology, University of Warsaw, 00-183 Warsaw, Poland; aqq@psych.uw.edu.pl; 5Department of Biochemistry and Molecular Biology, Centre of Postgraduate Medical Education, 01-004 Warsaw, Poland; dorota.sys@cmkp.edu.pl; 61st Department of Obstetrics and Gynecology, Centre of Postgraduate Medical Education, 01-004 Warsaw, Poland; anna.scholz@cmkp.edu.pl; 7Center for Midwifery and Women’s Health Research, Faculty of Health and Environmental Sciences, Auckland University of Technology, Auckland 1010, New Zealand; susan.crowther@aut.ac.nz

**Keywords:** intrapartum care, foetal position, right and left foetal position, difficulty of labour, Poland

## Abstract

Background: Many studies have shown the negative influence of the foetus’s occiput posterior position during birth on the final perinatal outcome. This study aims to add to the discussion on the impact of foetus positioning on the course of labour and subjective assessment of the level of labour difficulty. Methods: The cross-sectional study took place from February 2020 to September 2021, and consisted of filling out observation forms and the assessment by the midwives and women of the level of labour difficulty. This study is based on the observation of 152 labours in low-risk women. Findings: When compared to left foetal positioning, labours in which the foetus was in the right position were longer and more frequently failed to progress (in 11.3% vs. 37.5%), and epidural was more frequently administrated (in 30.4% vs. 52.7%). Both women and midwives subjectively evaluated deliveries with a foetus in the right position as more difficult. Conclusions: The right positioning of the foetus was related to greater labour difficulty and worse perinatal outcomes. The position of the foetus’ head in relation to the pelvis should be considered as an indicator of the difficulty of labour and a support plan for the woman should be offered accordingly.

## 1. Introduction

The term optimal foetal positioning (OFP) was introduced to the obstetrics and midwifery community by Jean Sutton [1], a New Zealand midwife and experienced childbirth educator. Sutton deduced from her many years of clinical observations that the positioning of the foetus in the uterus and its positioning in relation to the pelvis have a significant effect on childbirth. She defined OFP as when the foetus is head down, left position with its back facing the front of the uterus (LOA). As suggested, a decrease in the occurrence of this position of the foetus, as observed in recent years, could be linked to changes in the lifestyle and activities of pregnant women [1,2]. In response to the preference given to OFP, an approach called Spinning Babies^®^ [3] emerged. It uses gravity, balance, and movement to achieve LOA position of the foetus at the end of pregnancy or during labour that is not progressing as expected. Within this approach, workshops for parents and those in the obstetric and midwifery community were developed. Despite its popularity, the legitimacy and efficacy of this method have not been verified by research. Some studies questioned left occiput anterior (LOA) as the optimal foetal position (OFP) and suggest that prenatal practices encouraging this position are unnecessary [4].

A different approach to foetal positioning is based on the assumption that each pelvis has a specific shape that cannot be categorised by the four types described by Caldwell-Moloy [5]. Therefore, for each particular pelvis, there is a different OFP and the right position for some pelvises cannot be considered suboptimal. This approach suggests that instead of trying to attain a given foetal position, focus should be on minimalizing negative results during childbirth by measures taken to assist in delivering a foetus in the right position rather than striving to attain a given foetal position [6]. This approach also has not been verified by research.

Although it has not been proven that an OFP exists [4], some foetal positions are related to the greater number of medical interventions and complications during delivery. Most studies have focused on births during which the foetus is in the posterior or anterior position [7,8,9]. Less is known how the right or left positioning of the foetus at the onset of labour impacts the birth process, perinatal and neonatal outcomes, and subjective assessment of labour difficulty by women and care providers. 

This study aims to add to the discussion on the impact of foetus positioning on the course of the labour by comparing the perinatal and neonatal outcomes and subjective assessment of the level of labour difficulty by women and midwives in deliveries with right and left foetal positions. Particularly, it seeks to answer the question as to whether right or left foetal positioning diagnosed at the moment the foetal head drops down into the pelvis could be used as a predictor of labour difficulty that could inform care for women who may need an extra intrapartum support. 

## 2. Methods

The pilot cross-sectional study took place from 1 February 2020 to 30 September 2021 in the tertiary St. Sophia Specialist Hospital in Warsaw, which oversees approximately six thousand deliveries annually (including approximately 400 in the midwifery-led Hospital Birth Centre). In 2020, the hospital reported a caesarean rate of 34.2% (according to personal communication with the hospital). In the same year, the total caesarean rate for Poland was 45.1% [10]. 

Observations of labours were performed by midwives, including the first author (MW), working in the labour unit and midwifery-led Hospital Birth Centre at St. Sophia Specialist Hospital. Midwives from both units were invited to participate in the study, and 31 midwives agreed. They were between 24 and 54 years of age (Me = 44) and had between 3 and 32 years (Me = 22) of working experience. Due to the very high number of labours taking place in our hospital, up to 600 per month, and the procedure of vaginal examination in labour at least every 2 h, even midwives with less seniority had relevant experience. Those who agreed to participate were provided with information on the study and trained on how to record data in the observation forms used (the labour observation form and assessment of the level of labour difficulty form) (see Appendix A). 

Both forms used in the study to record data (the labour observation form and assessment of labour difficulty form) were designed using the Delphi method. The expert panel consisted of maternity care providers with many years of experience working in the delivery room. After completing the pilot, some questions on the labour observation form were modified to obtain precise and reliable information. Forms were designed and used in Polish and then translated into English by the third author for the purpose of this article. 

The labour observation form collected information on the position of the placenta, visible abnormalities of the pelvis, contractions, duration of stages of labour, failure to progress, foetal head rotation, bleeding, perineal injury, epidural, how the birth ended (vaginal, instrumental, or caesarean section), and Apgar score (in 5th minute). The form assessing the labour difficulty recorded midwife’s and woman’s independent assessment of labour difficulty using a scale from 1 to 10, where 1 referred to minimal and 10 to maximum difficulty. The assessment was carried out no later than 5 h following delivery.

The observations were conducted during births that took place in a standard delivery room (n = 105) and in the midwifery-led Hospital Birth Centre located on the hospital grounds (n = 33). As some of the midwives participating in the study also attended home births, we decided to include those observations (n = 14). The midwives conducting the observations were the primary care providers during these deliveries. The observations were made during their planned hospital shifts and planned homebirths. Labouring women were asked for a permission to include observations of their births in the study. They were provided with written information on the study and signed an informed consent form. 

The observations were made during births that met the following inclusion criteria: women giving birth over 18 years old, single pregnancy at term (between the 37 and 42nd week of gestation) and a recent (no older than 1 week) ultrasound scan describing the position of both the foetus and the placenta. Criteria for exclusion were as follows: women’s refusal to participate in the study, having indications for the induction of labour for reasons other than post-date pregnancy (for example: age of mother, low amniotic fluid index, large foetus, pregnancy after in vitro fertilisation, previous caesarean section, poor foetal movements, and/or a combination of these indications). Conditions in women such as diabetes, cholestasis, high blood pressure, and diagnosed prenatal illness were further criteria for exclusion as they could negatively influence labour. All the women recruited were included in the final analysis. 

Women who laboured at the hospital had an ultrasound scan when admitted to the labour ward. Only women who laboured at home did not have an ultrasound scan at the beginning of labour. In this case, an ultrasound scan no older than one week was required. At the beginning of the observation, the midwife determined the positioning of the foetus using Leopold’s manoeuvres. The position of the head toward the birth canal was considered the initial position. During labour, the midwife observed cranial sutures and the position of the fontanelle. Intrapartum ultrasound scans were not conducted in order to ensure women’s comfort. The foetal position was described according to the scheme in Figure 1. For some groups (LOA and ROP), due to the very small number of women, the statistical analysis is biased by significant measurement error. 

The latent phase of labour was defined in this study as the moment from the first irregular contractions to cervical dilation of 4 cm. The first stage of labour was defined as the moment when regular contractions lead to labour progress. The second stage of labour was measured when the cervix was fully dilated, as verified by digital examination. The physiological blood loss after vaginal childbirth was defined as no more than 350 mL. 

The women were divided into two groups based on the foetal position: the left position group (group I) and right position group (group II). STROBE reporting criteria were applied (Clinical Trial Registry and Registration number: PN/29/2020).

For statistical analysis, we represented continuous measurements as means (SDs) compared with Student’s *t*-test. Levene’s test was used to assess the homogeneity of variance. The Mann–Whitney U-test was employed only to analyze the differences between the compared groups in terms of assessing the level of labour difficulty. Categorical variables were expressed as numbers (%) and compared by the χ^2^ test. The statistical significance was 0.05. The calculations were made using the Statistica 13.1 program. 

The minimum size of the groups was estimated using the SPSS package 29.0.0.0(214) version. On the basis of WHO recommendations regarding intrapartum care for a positive childbirth experience [11], we assumed that the differences in the duration of individual stages of labour measured by standard deviation were within the following ranges: latent phase—SD 90–192 min, phase I—84–396 min, and phase II—5–54 min. For the purposes of the estimation, we adopted the least optimistic, i.e., the highest, standard deviations observed in the studies. In addition, the first author, who has over 25 years of clinical experience working on the labour ward, estimated that the mean differences observed by her in the duration of particular stages of labour between women with a foetus in the right and left position were as follows: for primiparous women approximately 600 min for the latent phase, 300 min for phase I and 60 min for phase II; for multiparous women, approximately 120 min for the latent phase, 180 min for phase I, and 20 min for phase II. Taking the approximate ratio of primiparous to multiparous women to be 1:1 (data obtained from St. Sophia’s Hospital for 2010–2020), we calculated the mean values of the estimated differences; these were 360 min for the latent phase, 240 min for phase I, and 40 min for phase II. We determined the minimum sizes of the compared groups on the basis of these calculations, assuming statistical significance at the level of 0.05 and the test power at the level of min. 0.8. The results were N ≥ 5 for the latent phase, N ≥ 35 for phase I and N ≥ 24 for phase II. We decided the compared groups should be larger than 70 people to take into account the possible imprecision of estimates and the risk that some of the participants may, at various stages of the study, drop out of the analysis. A total of 81 women with the right foetal position and a comparable number (71) with the left foetal position were therefore recruited for the study.

In the “post hoc” analysis, assuming that the observed distributions of the variables reflect the distribution in the general population (by estimating the standard deviations of the population through the deviations observed in the studied samples), we estimated the power of Student’s *t*-test for individual stages of labour for the sample included in the study. The results indicated the high power of the test for three out of four variables and unsatisfactory power for the duration of the first phase of labour. Low power for this variable resulted from a large variation in the observed times for women with the right foetal position.

## 3. Results

In total, 152 births were observed: 71 women were diagnosed with left foetal positioning (group I) and 81 with right foetal positioning (group II). The women were between 20 and 46 years of age (mean 31.4, SD 4.7), BMI 15.6–38.4 (mean 22.3, SD 3.3), and newborn birthweight varied from 2450 g to 4600 g (mean 3503.5, SD 4.7). Some 102 were primigravidas, and 50 were multigravidas. There were no significant differences noted between the two groups regarding age, parity, height, weight, BMI, or birthweight. However, there was a more significant increase in weight during pregnancy among women in group II (right foetal positioning), at 13 vs. 14.99 (*p* = 0.009) (Table 1). 

The distribution of individual initial positions of the head in the right foetal position was as follows: ROP 45.1%, ROT 35.4%, ROA 6.15%, and a position difficult to determine 13.35%. In the left position, it was as follows: LOA 28.6%, LOT 57.1%, LOP 4.3%, and a position difficult to determine 10%.

The right foetal position was more frequent in women with placenta positioned on the front wall of the uterus, at 39.4% (n = 26) vs. 61.3% (n = 46), χ^2^(1) = 6.762; *p* = 0.009. Women with right foetal positioning had also a significantly higher rate of labour induction, at 12.7% (n = 9) vs. 25.9% (n = 21), χ^2^(1) = 4.192; *p* = 0.041.

Discoordination of contractions occurred more frequently in group II (right foetal positioning) at 14.1% (n = 10) vs. 25.6% (n = 38), χ^2^(1) = 18.837; *p* < 0.001. Normative contraction activity was more common in group I (left foetal positioning) at 63% (n = 45) vs. 34.1% (n = 19), χ^2^(1) = 24.739; *p* < 0.001.

The durations of the individual phases of labour were longer in the group of women with right foetal positioning. This pertains to both the latent and active phases of the first stage of labour and the resting and active pushing phases of the second stage (Table 2).

When the examination of the cranial sutures and the placement of the fontanelle were considered, a more significant difference in the duration of stages of labour was observed (Table 3). 

Failure of labour progression was also more frequently observed in group II (right foetal positioning). In the first stage of labour, failure of progression was noted in 11.3% (n = 8) vs. 37.5% (n = 30) χ^2^(1) = 13.745; *p* < 0.001, and in the second stage in 4.4% (n = 3) vs. 25.4% (n = 18) χ^2^(1) = 11 *p* < 0.001. More frequent use of epidural was noted among women in group II 30.4% (n = 21) vs. 52.7% (n = 39) χ^2^(1) = 7.270; *p* = 0.007. A total of 27% of the participants after epidural experienced a change in positioning and rotation of the head—these findings did not differ among groups. 

The rate of caesarean section was also higher among participants with right foetal positioning 7% (n = 5) vs. 24.7% (n = 20) χ^2^(1) = 8.576; *p* = 0.003. Large perineal lacerations were more frequent among women with right foetal positioning 10.4% (n = 7) vs. 25.4% (n = 17), whereas small perineal lacerations were more frequent among women with left foetal positioning 64.2% (n = 43) vs. 46.3% (n = 31) φ-Yule = 0.216; *p* = 0.045.

Larger blood loss (above 350 mL) was more frequent among women with right foetal positioning compared to women with left foetal positioning 19.7% (n = 13) vs. 36.4% (n = 24) χ^2^(1) = 4.544; *p* = 0.033. Analysing the angle of head rotation to the final position of DOA, foetuses from right positioning made a greater turn than those from left positioning. No significant differences were noted between the groups in relation to the moment in which the rotation took place, i.e., during the first or second stage of labour. Only six women gave birth in the DOP position, of which five had a foetus in the right position and one had a foetus in the left position. During one birth, the foetus rotated from ROP to DOP and was delivered via caesarean section. This was an isolated incident so no conclusions could be drawn. A larger sample is needed for continued observation. 

The study did not show any significant differences regarding the Apgar score of the newborns or the number of transfers to the intensive neonatal care units (only one child in right foetal positioning group was transferred). Only six children scored less than 10 Apgar points (four in the right foetal positioning group and two in the left positioning group), and in the right group, four children scored less than 10 Apgar points. None of the children scored less than 8 Apgar points at the tenth minute.

The assessment of labour difficulty by the women and midwives showed a significant correlation. Both groups on a scale of 1–10 (where 1 referred to minimal difficulty and 10 to maximum difficulty) assessed births with right foetal positioning as more difficult (midwives: 6.13 vs. 4.44; U = 1510.5; *p* < 0.001; mothers 7.28 vs. 5.90; U = 1738; *p* < 0.001).

## 4. Discussion

The study showed significant differences in the course of labour and the perinatal outcomes for women with the right and left foetal positioning. The right foetal positioning was related to greater labour difficulties and worse perinatal outcomes such as prolonged labours, more significant perineal injuries, higher blood loss, and higher rates of medical interventions. No differences in neonatal outcomes were found. 

The differences found in our study were consistent with those described by studies looking at deliveries with foetal anterior (OA) posterior occipital position (OP) [8,12,13]. In our study, 45.1% of women who had right foetal positioning also had right occiput posterior position (ROP), compared to 4.3% of women all with left positioning who had left occiput posterior position (LOP). This reflects disproportions described in the literature indicating the more frequent appearance of ROP than LOP [14]. This may be linked to anatomical conditions such as the uterus tilted to the right, positioning of the bladder in the right frontal part of the pelvis and the rectum in the left rear part of the pelvis, or a dense liver on the right side of the foetus [14].

The higher rate of OP positioning among women with right foetal positioning may be linked to the greater labour difficulties found in this study for this group. When we separately compared anterior and posterior births for both right and left foetal positioning, labours with right positioning were still longer and more often failed to progress and ended with medical interventions. However, the sizes of the groups were too small to draw definitive conclusions. Thus, further studies on a larger sample that would allow comparisons between those groups are needed. 

In our study, right foetal positioning more often occurred among women with a placenta located on the front wall of the uterus. Previous studies have concluded that the localisation of the placenta may influence the foetal position in the uterus (cephalic versus breech and posterior versus anterior) and the anterior location of the placenta is linked to worse perinatal outcomes and higher rates of labour induction [15,16,17,18]. As in the case of anterior and posterior occiput position, this may have influenced the differences found between right and left positioning in our study group. However, when we compared induction rates only between women with anterior placenta, they were still higher among women with right foetal positioning. This would suggest that it is not only the anterior position of the placenta but also the right positioning of the foetus that increases the risk of induction. Again, studies on a larger sample are needed.

Furthermore, this study showed that discoordination of uterine contractions was more common in labours with right foetal positioning. Research by Buhimschi et al. indicated no differences in uterine contractility between posterior occiput and anterior occiput groups during either the first or the second stage of labour [19]. This suggests that the discoordination observed in our study was related to the right positioning of the foetus. Further studies on the possible mechanism of this discoordination are needed, particularly studies assessing whether there are differences in the surface of contact between the head and the cervix when the foetal head flexes in the right and left position. As the contact between the head and the cervix stimulates contractions (the Ferguson reflex) [20], the differences in the surface in contact could explain the dysfunction of uterine contractions described in our study. 

Both, midwives and women in our study assessed births with right foetal positioning as more difficult. In the case of midwives, this higher level of difficulty could be linked to the greater support required by women and the need to use extra midwifery skills to aid the progression of labour. In the case of women, studies showed that many factors influence women’s childbirth experience, including the duration of labour, use of an oxytocin drip, perinatal injury, and the mode of delivery [21,22,23,24]. In our study, for women with right foetal positioning, all those aspects were unfavourable. This may also explain the higher rate of epidural use in this group. Studies show that duration of labour and medical interventions, particularly the use of oxytocin drip, increase women’s use of epidural [25].

Clinicians differentiate between physiologically normal OP at the beginning of labour and pathologically persistent OP positions that remain OP until delivery [2]. Most studies focus on the latter [8,12], and little is known about how initially suboptimal foetal position may influence the course of birth. In our study, irrespective of their right or left initial positioning, most foetuses rotated during the second stage of labour (most often in the mid-pelvis) and ended up in a DOA position. Thus, clinically, they would be considered within the norm. Yet, as indicated by our study, the course of these labours differed. Although, as explained above, due to the relatively small groups in our sample, we were not able to conclusively show that the difference resulted from the right or left positioning, initial right foetal positioning seems to be a good predictor of labour difficulties. 

However, considering the limited evidence of the effectiveness of manoeuvres to achieve OFP and the inconclusiveness of existing research on the effectiveness of manual rotation [12,13,26,27,28,29,30,31,32,33], we believe initial suboptimal foetal position should not lead to an immediate intervention. Instead, our knowledge should be used to tailor appropriate care for labouring women. In fact, in our study, OFP as defined by Sutton [1] was very rare. Hasty choices at the beginning of labour may lead to increased incidence of operative deliveries without improving outcomes [34,35]. 

That said, Guittier et al. [36] showed that some of the manoeuvres aimed at optimising foetal position, although ineffective in changing the position of the foetus, had a positive effect on labouring women [36]. More research is needed to address maternal positions during labour, among other techniques, as this seems to be of key importance in tailoring care for women with unfavourable foetal positioning.

It is important to acknowledge a regional variation in pelvic morphology and the way the shape of the pelvis influences the process of birth. The way the foetus drops into the pelvis is at times dictated by the shape of the pelvic brim [37]. In an anthropoid pelvis that is narrow and deep, the shape of the brim harmonises with the foetus in the ROP position [7]. In this case, any kind of activity aiming to change the foetal position is unjustified [7]. Thus, foetal positioning in LOP, ROP, or DOP may be optimal for certain types of pelvis. However, this does not necessarily mean the labour will not be prolonged or more difficult. LOP, ROP, or DOP positions should not be treated as non-physiological, nor should they be corrected before or during labour [20]. Research shows that in the majority of cases, the foetus rotates clockwise [33,38]. Thus, it can be easier to reach proper rotation to the final position, DOA, in right foetal positioning (ROP) than in the case of left foetal positioning (LOP). 

Based on our findings, we suggest that optimal labour care should be developed to screen for suboptimal foetal positions, especially the right occiput posterior position (ROP), and to minimalize potentially negative consequences. Our research shows that women with ROP may require more attentive care and access to wide range of non-pharmacological and pharmacological (including epidurals) methods to alleviate pain as well as greater support from birth partners. Furthermore, during labour, techniques that allow for full mobility of the pelvis should be used so that the internal spaces of the pelvis can be mechanically increased due to the biodynamics of the birth canal and the birth process [39,40,41]. 

## 5. Limitation 

Due to the limited number of midwives participating in the study, not all women who met the inclusion criteria and who gave birth during the study period in the hospital where the research was conducted could be included in the study. This could have led to selective bias. The sample size was relatively small, particularly in the case of rare positions such as ROA or LOP. This was due to the rare occurrence of these positions in the general population. Due to the small number of women in these groups, we were unable to account for the impact of posterior position. It was also not always possible to fully observe cranial sutures during childbirth. Due to the limited number of internal examinations and their frequency, it was difficult to precisely determine in which pelvic space the head rotated. Furthermore, the evaluation of the difficulty of labour by the woman is a tool that brings a subjective assessment. Often, multigravidas compare the current labour to previous ones. Therefore, our findings need to be generalised with caution, and our study is considered a pilot study. However, it is important to note that further research is needed to confirm these pilot findings and to explore potential confounding factors.

## 6. Conclusions 

Right foetal positioning was related to greater difficulty during labour and worse perinatal outcomes. Therefore, considering the initial position of the head as it drops into the pelvis is essential while foreseeing delivery difficulties. In turn, appropriate support for women should be planned and organised. 

## Figures and Tables

**Figure 1 healthcare-12-00864-f001:**
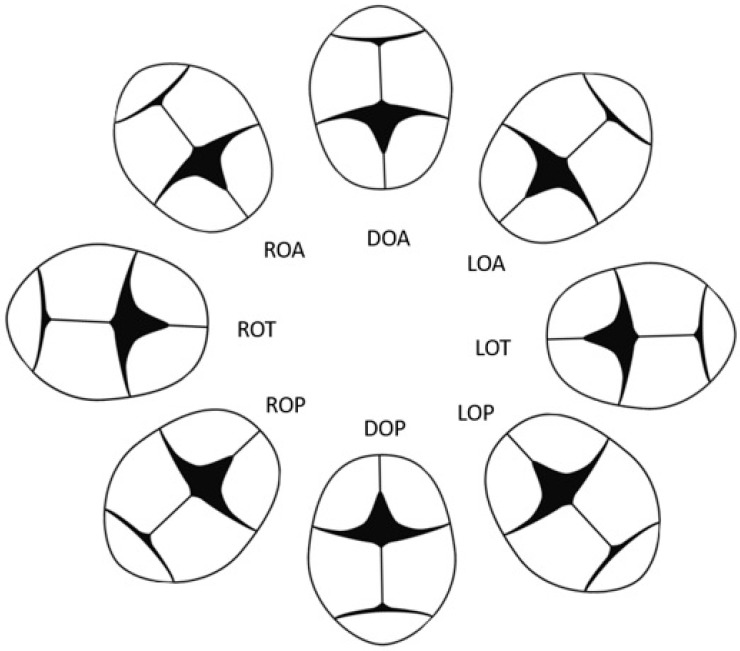
Variations of the foetal head position (DOA, direct occipito-anterior; DOP, direct occipito-posterior; LOA, left occipito-anterior; LOT, left occipito-lateral; LOP, left occipito-posterior; ROA, right occipito-anterior; ROT, right occipito-lateral; ROP, right occipito-posterior).

**Table 1 healthcare-12-00864-t001:** Characteristics in groups with right and left foetal positioning.

Variable	Foetal Position	N	Mean	SD	SEM	95%CI		Significance of the Difference
						Lower Limit	Upper Limit	
Age	left	71	31.11	4.73	0.56	30.01	32.21	t(150) = 0.671; *p* = 0.503
	right	81	31.63	4.75	0.53	30.60	32.66	
Week of pregnancy	left	71	39.58	1.02	0.12	39.34	39.82	t(150) = 1.334; *p* = 0.184
	right	81	39.81	1.15	0.13	39.56	40.07	
Height	left	71	167.32	5.46	0.65	166.05	168.59	t(150) = 1.504; *p* = 0.135
	right	81	168.77	6.26	0.70	167.40	170.13	
Weight before pregnancy	left	71	62.70	11.11	1.32	60.12	65.29	t(150) = 0.417; *p* = 0.672
	right	81	63.38	8.59	0.95	61.51	65.25	
BMI	left	71	22.40	3.87	0.46	21.50	23.30	t(124) = 0.276; *p* = 0.783
	right	81	22.25	2.73	0.30	21.65	22.84	
Weight gained	left	71	13.00	4.77	0.57	11.89	14.11	t(150) = 2.660; *p* = 0.009
	right	81	14.99	4.44	0.49	14.02	15.95	
Newborn birthweight	left	71	3441.20	362.02	42.96	3357.00	3525.40	t(150) = 1.874; *p* = 0.063
	right	81	3558.02	401.15	44.57	3470.66	3645.38	

**Table 2 healthcare-12-00864-t002:** Duration of stages of labour (in minutes) in groups with left and right foetal positioning.

	Foetal Position	N	Mean	SD	SEM	Limits of Confidence Intervals (95% CI)		Levene’s Test Equality of Variances	Student’s *t*-Test for Independent Samples (Directional Hypothesis)
Latent phase	left	69	305.62	389.74	46.92	213.66	397.59	F = 4.209; *p* = 0.042	t(140 *) = 3.919; *p* < 0.001
	right	74	582.31	453.92	52.77	478.89	685.73		
Phase I	left	69	333.32	166.65	20.06	294.00	372.64	F = 2.759; *p* = 0.099	t(141) = 1.816; *p* = 0.036
	right	74	506.85	776.99	90.32	329.82	683.88		
Phase II	left	68	37.94	28.42	3.45	31.19	44.70	F = 27.759; *p* < 0.001	t(110 *) = 5.449; *p* < 0.001
	right	70	75.49	49.92	5.97	63.79	87.18		
Active pushing	left	68	27.38	17.00	2.06	23.34	31.42	F = 4.209; *p* = 0.042	t(140) = 3.919; *p* < 0.001
	right	69	43.52	31.48	3.79	36.09	50.95		

* Due to the significant result from Levene’s test, the heterogeneity of variance in the two groups was taken into consideration.

**Table 3 healthcare-12-00864-t003:** Duration of stages of labour (in minutes) in groups with left and right foetal positioning including the examination of the cranial sutures and the placement of the fontanelle.

	Position at the Onset of Stage I	N **	Mean	Standard Deviation	Standard Error of the Mean	1.96	Limits of Confidence Intervals (95% CI)	
**Latent phase**	LOA	19 (21)	**189.63**	154.57	35.46	69.50	**120.13**	**259.13**
	ROA *	6 (8)	**475.00**	215.01	87.78	172.05	**302.95**	**647.05**
	LOT	40 (41)	**301.38**	259.38	41.01	80.38	**221.00**	**381.76**
	ROT	23 (29)	**568.30**	438.48	91.43	179.20	**389.10**	**747.50**
	LOP *	5 (5)	**714.00**	1213.87	542.86	1064.01	**0.00**	**1778.01**
	ROP	32 (39)	**536.88**	354.34	62.64	122.77	**414.11**	**659.65**
**Phase I**	LOA	19 (21)	**318.79**	154.92	37.84	74.17	**244.62**	**392.96**
	ROA *	6 (8)	**555.83**	302.78	123.61	242.28	**313.55**	**798.11**
	LOT	40 (41)	**335.18**	162.62	25.71	50.39	**284.79**	**385.57**
	ROT	23 (29)	**421.43**	1660.97	33.56	65.78	**355.65**	**487.21**
	LOP *	5 (5)	**376.00**	205.77	92.02	180.36	**195.64**	**556.36**
	ROP	32 (39)	**422.66**	261.14	46.16	90.47	**332.19**	**513.13**
**Phase II**	LOA	19 (21)	**38.79**	27.16	6.23	12.21	**26.58**	**51.00**
	ROA *	6 (8)	**58.00**	36.62	14.95	29.30	**28.70**	**87.30**
	LOT	40 (41)	**38.23**	30.20	4.77	9.35	**28.88**	**47.58**
	ROT	23 (29)	**70.74**	55.58	11.59	22.72	**48.02**	**93.46**
	LOP*	5 (5)	**29.80**	30.43	13.61	26.68	**3.12**	**56.48**
	ROP	32 (39)	**83.22**	47.89	8.47	16.60	**66.62**	**99.82**
**Active pushing**	LOA	19 (21)	**26.84**	15.32	3.52	6.90	**19.94**	**33.74**
	ROA *	6 (8)	**38.50**	27.74	11.32	22.19	**16.31**	**60.69**
	LOT	40 (41)	**28.13**	17.18	2.72	5.33	**22.80**	**33.46**
	ROT	23 (29)	**40.91**	26.78	5.58	10.94	**29.97**	**51.85**
	LOP *	5 (5)	**20.20**	23.03	10.30	20.19	**0.01**	**40.39**
	ROP	32 (39)	**48.34**	34.25	6.05	11.86	**36.48**	**60.20**

* small group; results are biased by significant measurement error. ** the number of women with this positioning included in the study is provided in parentheses. Only those who had complete data in the analysed variables were included in the statistical analysis (for example, women who had a caesarean section were excluded).

## Data Availability

Data are contained within the article.

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
