# Peer review of "Perinatal Outcomes and Level of Labour Difficulty in Deliveries with Right and Left Foetal Position—A Preliminary Study"

_healthcare, 2024, doi:10.3390/healthcare12080864_

Round 1

Reviewer 1 Report

Comments and Suggestions for Authors

The authors present a study analyzing the influence of fetal position (right or left) on the course of labor and perinatal outcomes. The study topic is indeed intriguing. However, it is pertinent to address the following issues:

MAJOR POINTS

  1. The ethical committee endorsement, essential for an observational study, is not mentioned.
  2. The authors solely conduct an inferential comparative study between both cohorts, without determining through multivariate analysis the influence of various other factors that could potentially affect the course of pregnancy (fetal weight, maternal age, parity, etc.). This is crucial for understanding the true impact of fetal position and consequently for extrapolating conclusions.

MINOR POINTS

  1. References do not adhere to the journal's guidelines, which require numerical referencing in the text.
  2. The authors introduce the acronym OFP for Optimal Foetal Positioning but fail to abbreviate it in subsequent mentions.
  3. Fetal pH at birth, a marker of morbidity and a definitive indicator of intrapartum fetal distress, has not been included.
  4. The authors state: "The differences between the compared groups were analyzed using the Mann-Whitney U-test". I understand this applies only to non-parametric variables.
  5. I recommend replacing "average" with "mean".
  6. Tables are challenging to interpret in their current format; I suggest swapping rows and columns.
  7. The authors mention: "The study did not show any significant differences regarding Apgar score of the newborns or the number of transfers to the intensive neonatal care units," but fail to provide the actual data.
  8. Typo in line 247: "p=0,00", digits are missing.
Comments on the Quality of English Language

The authors present a study analyzing the influence of fetal position (right or left) on the course of labor and perinatal outcomes. The study topic is indeed intriguing. However, it is pertinent to address the following issues:

MAJOR POINTS

  1. The ethical committee endorsement, essential for an observational study, is not mentioned.
  2. The authors solely conduct an inferential comparative study between both cohorts, without determining through multivariate analysis the influence of various other factors that could potentially affect the course of pregnancy (fetal weight, maternal age, parity, etc.). This is crucial for understanding the true impact of fetal position and consequently for extrapolating conclusions.

MINOR POINTS

  1. References do not adhere to the journal's guidelines, which require numerical referencing in the text.
  2. The authors introduce the acronym OFP for Optimal Foetal Positioning but fail to abbreviate it in subsequent mentions.
  3. Fetal pH at birth, a marker of morbidity and a definitive indicator of intrapartum fetal distress, has not been included.
  4. The authors state: "The differences between the compared groups were analyzed using the Mann-Whitney U-test". I understand this applies only to non-parametric variables.
  5. I recommend replacing "average" with "mean".
  6. Tables are challenging to interpret in their current format; I suggest swapping rows and columns.
  7. The authors mention: "The study did not show any significant differences regarding Apgar score of the newborns or the number of transfers to the intensive neonatal care units," but fail to provide the actual data.
  8. Typo in line 247: "p=0,00", digits are missing.

Author Response

Dear Editor and Reviewers, We are very grateful for giving us the opportunity to improve the manuscript. We found all suggestions to be both thoughtful and helpful, and we believe the manuscript is improved as a result. Please, take a look at the attached file. 

Reviewer 2 Report

Comments and Suggestions for Authors

This pilot study investigates the effect of foetal head position during labour on the course of labour and on subjective perceptions of the difficulty of labour.

Overall, knowledge of the baby's position during labour is essential to ensure the safety and well-being of both mother and baby and to make informed decisions about the management of labour. Certain fetal positions are correlated with increased medical interventions and complications during labour. However, the emphasis on posterior or anterior positions neglects the potential influence of right or left fetal position at the onset of labour, which this study aimed to address.

This study seeks to fill a critical gap in existing research by examining the influence of right and left fetal position on perinatal and neonatal outcomes, and women's and care providers' subjective ratings of the difficulty of birth. By focusing on fetal position diagnosed at the time of fetal head descent into the pelvis, it seeks to identify potential predictors of birth difficulty that could inform specific intrapartum interventions.

Although the pilot study provides valuable insights into the relationship between fetal position and difficulties in childbirth, its methodological limitations require careful consideration. Problems with the limited number of participants, small sample size, difficulties in observation, and subjective assessment of birth difficulties underscore the need for further research with better methodology.

Please add the following: How was the quality of observation and examination during labour ensured for the midwives involved in the research? How many midwives were there involved? What was their length of practice, etc.?

The conclusions drawn from the study appear to be consistent with the evidence and arguments presented, which address the central issue of the effect of fetal head position at birth on the course of labour and on the subjective assessment of the difficulty of labour. By observing 152 deliveries in low-risk women, analysing perinatal and neonatal factors, the study effectively answered the main question posed regarding the effect of fetal position on birth outcomes.

The study used a cross-sectional design and involved the completion of observation forms and ratings by midwives and women regarding the level of difficulty of delivery. By comparing outcomes between deliveries with fetuses in the left and right positions, it provided concrete evidence to support its conclusions. The results of the study support the conclusion that right fetal position is associated with greater birth difficulty and poorer perinatal outcomes. The recommendation to consider the position of the fetal head relative to the pelvis as an indicator of the difficulty of delivery and to adjust support plans accordingly is consistent with the evidence presented in the study. However, it is important to note that further research is needed to confirm these pilot findings and to explore potential confounding factors.

Citations are not uniformly given. An adjustment is needed. For example:

Line 325 Popowski, Porcher, Fort, Javoise, & Rozenberg, 2015 (not et al.)

Line 329 M. J. Guittier et al., 2016

Line 44 Elmore, McBroom, & Ellis, 2020 and Line 306 Elmore et al., 2020).

Table 2 shows the different numbers of women with left and right fetal position during the different stages of labour. Please explain the reasons.

Author Response

(The authors gave the same response as above.)
